# The Purinergic P2X7 Receptor as a Target for Adjunctive Treatment for Drug-Refractory Epilepsy

**DOI:** 10.3390/ijms25136894

**Published:** 2024-06-23

**Authors:** Divyeshz Thakku Sivakumar, Krishi Jain, Noura Alfehaid, Yitao Wang, Xinchen Teng, Wolfgang Fischer, Tobias Engel

**Affiliations:** 1Department of Physiology and Medical Physics, Royal College of Surgeons in Ireland, University of Medicine and Health Sciences, D02 YN77 Dublin, Ireland; divyeshzthakkus21@rcsi.ie (D.T.S.); krishijain22@rcsi.ie (K.J.); nouraalfehaid20@rcsi.ie (N.A.); yitaowang23@rcsi.ie (Y.W.); 2International College of Pharmaceutical Innovation, Soochow University, Suzhou 215123, China; xcteng@suda.edu.cn; 3Independent Researcher, 04275 Leipzig, Germany; wefis@gmx.de; 4FutureNeuro, Science Foundation Ireland Research Centre for Chronic and Rare Neurological Diseases, Royal College of Surgeons in Ireland, University of Medicine and Health Sciences, D02 YN77 Dublin, Ireland

**Keywords:** epilepsy, seizures, drug-refractoriness, purinergic signalling, P2X7 receptor, adjunctive treatment

## Abstract

Epilepsy is one of the most common neurological diseases worldwide. Anti-seizure medications (ASMs) with anticonvulsants remain the mainstay of epilepsy treatment. Currently used ASMs are, however, ineffective to suppress seizures in about one third of all patients. Moreover, ASMs show no significant impact on the pathogenic mechanisms involved in epilepsy development or disease progression and may cause serious side-effects, highlighting the need for the identification of new drug targets for a more causal therapy. Compelling evidence has demonstrated a role for purinergic signalling, including the nucleotide adenosine 5′-triphosphate (ATP) during the generation of seizures and epilepsy. Consequently, drugs targeting specific ATP-gated purinergic receptors have been suggested as promising treatment options for epilepsy including the cationic P2X7 receptor (P27XR). P2X7R protein levels have been shown to be increased in the brain of experimental models of epilepsy and in the resected brain tissue of patients with epilepsy. Animal studies have provided evidence that P2X7R blocking can reduce the severity of acute seizures and the epileptic phenotype. The current review will provide a brief summary of recent key findings on P2X7R signalling during seizures and epilepsy focusing on the potential clinical use of treatments based on the P2X7R as an adjunctive therapeutic strategy for drug-refractory seizures and epilepsy.

## 1. Introduction

Epilepsy comprises a heterogeneous group of brain diseases characterised by the occurrence of recurrent spontaneous seizures [1,2]. With an incidence of 1–2% within the general population and affecting approximately 70 million people worldwide, epilepsy represents one of the most common chronic brain diseases, with incidences being higher among the young and the elderly [3]. Further reducing the quality of life of patients, epilepsy comes along with social disadvantages (e.g., unemployment, stigma), and the increased risks of premature mortality (up to 3-fold) and comorbidities (up to 8-fold) such as depression and anxiety [3,4,5,6]. The causes of epilepsy can be of a genetic nature (e.g., de novo mutations, polymorphisms) or due to a precipitating injury to the brain such as a stroke, traumatic brain injury (TBI), infections, tumours, or an episode of status epilepticus (SE). In adults, the most common form of epilepsy is temporal lobe epilepsy (TLE). TLE involves different structures within the limbic system, including the amygdala and hippocampus, and is normally accompanied by hippocampal sclerosis, characterised by selective neuronal loss and reactive gliosis [7]. Of note, TLE represents the most studied form of epilepsy in the setting of purinergic signalling [8]. Epileptogenesis, which usually follows a precipitating brain injury, is the process of turning a healthy brain into a brain experiencing recurrent seizures and involves a plethora of pathological processes including aberrant neuronal plasticity, changes in neurotransmitter release, blood–brain barrier (BBB) dysfunction, and inflammation among many others [9,10]. Notably, inflammatory processes have attracted particular attention over recent decades as potential sources for new drug targets [11].

Anti-seizure medications (ASMs), heavily focused on targeting synaptic transmission (e.g., potentiating of γ-aminobutyric acid [GABA]ergic inhibition or reduction of glutamatergic excitation) and blocking Na^+^ and/or Ca^2+^ channels, remain the first-line treatment for epilepsy. Other therapeutic strategies include resective surgery, neuromodulation devices (e.g., vagus nerve stimulation), and therapeutic diets (e.g., ketogenic diet) [12,13,14]. However, despite the availability of over 30 ASMs, epilepsy treatment remains symptomatic, primarily aimed at reducing the risk of seizure recurrence [3]. Critically, responder rates have changed little over recent decades, with approximately 30% of patients not responding to current ASMs, even if given as a combination treatment of different anticonvulsants [3,15]. Importantly, patients suffering from drug-resistant epilepsy, defined as a “failure of adequate trials of two tolerated, appropriately chosen, and used ASM schedules (whether as monotherapy or in combination) to achieve sustained seizure freedom” [16], are at a greater risk of developing injury, psychosocial dysfunction, early death, and a lower standard of living [5]. Multiple pathological changes have been identified that potentially contribute to drug refractoriness in epilepsy (e.g., alterations of the GABA_A_ and/or glutamate receptors, altered sensitivity of the Na^+^ and/or Ca^2+^ channels, the over-expression of multidrug transporter proteins at the level of the BBB, and the overexpression of efflux transporters in the peripheral organs, genetic variance, dysfunction of the BBB, structural alterations such as neurodegeneration and synaptic reorganisation, and gliosis and neuroinflammation [5,17,18,19,20]), which implies that efficient epilepsy treatment most likely requires the targeting of multiple processes rather than single pathological mechanisms. Further complicating epilepsy treatment, ASMs have no apparent impact upon the underlying causes and may exacerbate comorbidities, with drug-induced adverse side-effects being of particular concern (e.g., neurological and cognitive disturbances, fatigue, irritability, dizziness) [21]. There is, therefore, a persistent clinical demand for the identification of alternative drug targets effective in patients unresponsive to current ASMs.

Research over recent decades has provided compelling proof that purinergic signalling via the adenosine 5’-triphosphate (ATP)-gated cationic P2X7 receptor (P2X7R) is implicated not only in the generation of seizures (ictogenesis), but also epilepsy [8,22]. While several reviews published over recent years have provided a detailed summary of the data demonstrating a role for P2X7Rs during seizures and epilepsy (e.g., [8,22,23,24,25,26,27,28]), we here will focus on how drugs targeting the P2X7R may be used in a clinically relevant setting.

## 2. The P2X7 Receptor

Purinergic signalling is one of the most ancient and fundamental features of cellular signalling and neuron–glia interactions [29]. In a seminal study published in 1972, Geoffrey Burnstock suggested for the first time that extracellular ATP (eATP) acts as a co-transmitter in both the peripheral and central nervous system (PNS, CNS) [30]. The purinergic signalling system comprises enzymes, transporters, and receptors, including the ionotropic P2X and metabotropic P2Y receptor family, responsible for the synthesis, release, signalling transduction, and inactivation of eATP and its breakdown products (e.g., adenosine). It has been described to be functional throughout the body including the brain, where it has been shown to be involved in a plethora of cellular processes including neuro- and gliotransmission, neuromodulation, cellular survival, proliferation and differentiation, and neuroinflammation [31]. For a broader and more detailed introduction to purinergic signalling, please refer to other more extensive reviews (e.g., [31,32,33]).

The purinergic P2X7R (originally termed as the P2Z receptor [34]) belongs to the ionotropic P2XR family that consists of seven family members termed P2X1-7. The P2X7R is a 595 amino acid-long protein with two alpha-helical transmembrane domains termed TM1 and TM2, a large extracellular loop containing the three eATP-binding pockets, and intracellular N- and extended C-terminal tails [35,36,37]. A functional P2X7R comprises three subunits and, in contrast to other members of the P2XR family, does not form heteromeric assemblies with other members of the P2X family, although some interactions have been suggested with P2X4R [38,39]. Brief stimulation by its endogenous agonist eATP results in the rapid and reversible opening of a non-selective cation channel, mediating the rapid influx of Na^+^ and Ca^2+^ and efflux of K^+^, and other cations [40].

The P2X7R couples to multiple cell-specific intracellular signalling cascades that lead to the activation of diverse intracellular signalling molecules, such as phospholipases, caspases, tyrosine phosphatase, calcineurin, mitogen-activated protein kinase pathway proteins, and the NOD-, LPR-, and pyrin domain-containing protein 3 (NLRP3) inflammasome complex (for more details, see [41]). The P2X7R has some specific structural and functional characteristics that differentiate it from the other P2XR family members. This includes its ability to form macropores that are thought to be implicated in cell death mechanisms, prolonged activation and slower desensitisation kinetics, ectodomain shedding of cell-surface proteins, and a larger intracellular C-terminus which has been linked to pore formation and interactions with other proteins and intracellular signalling cascades (e.g., the extracellular signal-regulated kinases ([ERK]) pathway and activation of caspase-3) [42,43,44,45,46,47,48]. The N-terminus, in turn, has a specific interaction site for protein kinase C that is involved in the regulation of the Ca^2+^ flow through the channel, controlling the receptor gating and facilitating its activation [49]. Most notably, however, P2X7Rs have a 10-to-100-fold reduced sensitivity for eATP (EC_50_ ≥ 100 µM, activation threshold: 300–500 µM) compared to other P2XR subtypes [34]. Under physiological conditions, concentrations of eATP are within the range of nanomolar to low micromolar [50]. Thus, concentrations in the higher micromolar range required for P2X7R activation are suggested to occur mainly under pathological conditions of high ATP release, restricted to disease conditions (e.g., epilepsy, TBI) [51,52,53]. Consequently, it was proposed that eATP may act as a danger signal in the brain and P2X7Rs function as danger-sensors contributing to the progression of brain diseases [54]. On the other hand, regarding its wide expression in various mammalian cells, it can be suggested that this receptor may have a role in basal cell signalling in physiological functions and homeostasis [36]. Therefore, it is important to note that P2X7Rs, beside their tissue damage-activated actions, also carry out multiple functions during physiological conditions, which should be considered when evaluating the safety profiles of P2X7R-based therapies. This includes their effects on CNS development, synaptic plasticity and memory, neurogenesis, neurotransmitter release, and altered immune responses in P2X7R knock-out (KO) mice [55,56,57,58,59]. Moreover, further suggesting a physiological role of P2X7Rs in the CNS, we have shown altered mRNA and microRNA expression profiles during physiological conditions in hippocampal tissue from P2X7R KO mice [60,61,62]. It is, however, noteworthy to mention that P2X7R KO mice are viable and show protection against most noxious insults [63,64]. Importantly, as discussed within the following sections, P2X7R-based treatments may provide some degree of disease-modification [65,66], reducing the time of treatment and, thereby, the potential negative impact on normal brain function(s).

While there is a broad consensus of its presence in cells of the immune system such as microglia and macrophages, P2X7Rs have also been reported to be functional in other cell types in the brain including oligodendrocytes, endothelial cells, astrocytes, and neurons, with the latter remaining a matter of discussion [67,68,69]. New evidence suggests, however, that P2X7Rs are also functional in neurons in epilepsy [62], as discussed within the next section. The development of new and more accurate analytical tools (e.g., P2X7R-detecting nanobodies [70]) and new, more sophisticated in vivo models (e.g., cell type-specific P2X7R KO models [71]) will, without a doubt, shed further light on the specific expression and function of this receptor during both health and disease, which is critical for the design of the most effective treatment strategies.

To date, P2X7Rs have been implicated in numerous cellular and intracellular signalling cascades altered during seizures and epilepsy. Most notably, P2X7R activation promotes the formation of the NLRP3 inflammasome, leading in turn to the activation of caspase-1 and the subsequent release of the cytokine interleukin-1β (IL-1β) [72]. IL-1β is a well-established proinflammatory and proconvulsive cytokine, and drugs interfering with IL-1β signalling (e.g., anakinra) have shown promising anti-seizure effects [73,74,75]. The P2X7R has also been shown to contribute to the activation and proliferation of microglia [76,77], and reactive microglia, in turn, are recognised to contribute to seizures and epilepsy [78,79]. In addition to microglia, P2X7Rs have also been reported to be functional in astrocytes, where they possibly contribute to both inflammation and increased brain hyperexcitability [28]. While their ability to contribute to and promote inflammatory signalling represents the most studied pathway of how P2X7Rs contribute to diseases, it is important to bear in mind that P2X7R activation has been linked to a wide array of signalling pathways such as the control of cellular survival, synaptic transmission, BBB integrity, and aberrant plasticity [56,64,80,81,82,83]. As a consequence, the P2X7R has been linked to a plethora of brain diseases ranging from acute conditions such as a stroke to chronic conditions including neurodegenerative diseases, psychiatric conditions [63,64,84], and, as discussed within this review, epilepsy [24,25].

## 3. The P2X7R as a Treatment Target for Drug-Refractory Epilepsy

Research over recent decades has provided a large body of evidence demonstrating beyond doubt a causative role for the P2X7R during the generation of seizures (ictogenesis) and epilepsy. While early studies have focused on the analysis of changes in P2X7R expression and its cell-specific localisation following seizures and in epilepsy, more recent studies have investigated its contribution to seizure-induced pathology (e.g., seizures, neurodegeneration, neuroinflammation) and the anticonvulsive and antiepileptic potential of P2X7R antagonists [8,24]. Within the next paragraphs, following a brief overview of what is known regarding the release of its main endogenous agonist ATP during seizures, we will provide a summary of what is known regarding changes in P2X7R expression and function in epilepsy, discuss the most likely mechanisms of how P2X7Rs contribute to seizures and epilepsy, and how P2X7R-targeting drugs may be used in a clinical setting.

### 3.1. ATP Release during Seizures and Epilepsy

eATP is one of the primary damage-associated molecular pattern (DAMP) to be released at sites of tissue injury, serving as a physiological protective mechanism [54,85]. eATP concentrations are tightly regulated via different ATP release mechanisms and eATP-degrading ectoenzymes [86,87]. In the CNS, eATP has been described as a co-transmitter with glutamate [88,89] and GABA [90,91]. ATP has also been shown to be released from non-neuronal cells such as astrocytes [92]. ATP release mechanisms include exocytotic and non-exocytotic mechanisms involving the Cl^−^-dependent vesicular nucleotide transporter (VNUT) [93], voltage-dependent anion channels [94], ATP-binding cassette transporters [95], and the purinergic P2X7R and hemichannels such as connexins and pannexins [96,97,98]. Due to its huge concentration gradient between the intracellular and extracellular compartment, ATP has also been shown to be released passively through damaged cell membranes [54]. Once released, eATP is rapidly broken down via ectonucleotidases into different breakdown products (e.g., ADP, adenosine), signalling molecules in their own right [31]. Demonstrating altered concentrations during seizures, eATP concentrations have been found to be increased in the brain of a seizure-prone strain of mice [inbred DBA/2 (D2)] [99] and in slices of rat hippocampi stimulated via depolarising high K^+^ concentrations [100]. Using a rat model where SE was induced via an intraperitoneal injection of pilocarpine, Dona et al. reported increased extracellular levels of ADP, adenosine monophosphate (AMP), and adenosine. Purine levels, including eATP, were also found to be increased in the brain of mice following an epileptic seizure [51]. Regarding the ATP release mechanisms during seizures, most research has focused on the pannexin-1 hemichannel. Increases in eATP have been shown to be mediated via pannexin-1 in rat hippocampal slices [101]. More recent studies have shown that pharmacological inhibition of pannexin-1 channels reduced eATP concentrations in resected tissue from patients with epilepsy. Interestingly, pannexin-1 inhibition also blocked ictal discharges in human cortical brain tissue slices, and mice KO for pannexin-1 had a milder KA-induced SE phenotype [102], suggesting ATP release mechanisms as a potential target for seizure control. As previously mentioned, it is, however, important to keep in mind that, once released, eATP is rapidly broken down via the action of several ectonucleotidases into different breakdown products including adenosine [87]. Thus, while activating ATP-sensitive P2X receptors, eATP also contributes to the extracellular adenosine pool, thereby further impacting the seizure threshold, which will be briefly discussed in the next paragraph [103].

Adenosine is formed by the dephosphorylation of eATP from neurons and astrocytes via the membrane-bound ecto 5ʹ-nucleotidases CD39 and CD73, and transported through the lipid bilayer via nucleoside transporters in the extracellular space [86] where it is largely under the control of the enzyme adenosine kinase (ADK). Adenosine is now a well-characterised anti-inflammatory endogenous anticonvulsant of the brain and a mediator of seizure arrest [104,105,106,107]. A strong increase in extracellular adenosine during seizure activity in the hippocampi of patients was documented and considered to be sufficient to terminate ongoing seizure activity. On the other hand, the dysregulation of adenosine signalling (receptors, transporters, upregulation of ADK as well as DNA hypermethylation) is involved in the development of epilepsy [103]. Several studies have also reported changes in the density or affinity of the four adenosine receptors in animal models as well as in human epileptic tissues, but this needs further clarification. In animal models, the stimulation of the A1 (and A2A) receptor with brain-permeable agonists effectively contributes to seizure suppression. Of note, a systemic administration of adenosine itself is not possible due to strong cardiovascular side-effects [108]. In conclusion, treatments that facilitate the adenosine signalling system (adenosine augmentation therapies) emerge as a rational therapeutic target with the potential to suppress seizures, but also to prevent epileptogenesis [8]. Recently, intracerebral cell therapy approaches have been developed for long-term therapeutic adenosine delivery at sites of injury or pathology [8,103].

### 3.2. P2X7R Expression and Cell-Specific Localisation following Seizures and in Epilepsy

While there is wide agreement that P2X7R protein levels are increased in the brain following SE (rodents) and in epilepsy (rodents and patients), with studies mainly focused on the hippocampus and cortex [65,109,110,111,112,113,114], which cell types are involved remains to be fully determined. Increased P2X7R protein levels have been observed in both glial cells (microglia and oligodendrocytes) and neurons. However, the exact cell type-specific P2X7R profile seemed to be more dependent on the analytical tool used (e.g., type of antibodies, P2X7R reporter and KO mice, patch clamp) than the disease background or animal model. Using immunohistochemical approaches, P2X7Rs have been detected on glutamatergic nerve terminals and mossy fibres following intraperitoneal (i.p.) pilocarpine in epileptic rats [109,110]. P2X7Rs localised to neurons have also been observed following intra-amygdala kainic acid (IAKA) (SE and epilepsy) using a transgenic P2X7R reporter mouse model, which expresses a soluble enhanced green fluorescent protein (EGFP) driven by the *P2rx7* promoter [65,111]. These data have, however, been questioned due to the abnormal P2X7R expression and increased P2X4R expression in this reporter mouse [115]. In contrast, using a second P2X7R reporter mouse, where the P2X7R is fused to the fluorescent protein EGFP, these results could not be replicated and P2X7Rs seemed to be absent from neurons following IAKA in mice. The P2X7R protein levels were, however, shown to be increased in microglia and oligodendrocytes. No P2X7Rs were detected on astrocytes [68,114]. Using patch clamp recordings, increased P2X7R-dependent currents have also been detected in neural progenitor cells (NPCs) in the subgranular zone of the dentate gyrus in hippocampal brain slices [116]. Finally, in a recent study using the Cre-LoxP system, Alves et al. showed altered responses to chemoconvulsants in mice KO for the P2X7R in microglia (*P2rx7:Cx3cr1*-Cre) and neurons (*P2rx7*:*Thy-1*-Cre), suggesting that P2X7Rs are functional in both cell types. In the same study, the authors further showed increased P2X7R-mediated currents in GABAergic interneurons via a patch clamp in brain slices from mice subjected to IAKA and elevated *P2RX7* mRNA expression in excitatory and inhibitory neurons in the hippocampus of patients with TLE [62].

Taken together, while an increased P2X7R expression in microglia seems to be a common response to seizures, new evidence also suggests P2X7Rs to be increased in neurons. Future studies using, for instance, spatial transcriptomics/proteomics, electrophysiology (i.e., patch clamp in brain slices), or cell type-specific KO mice should, therefore, determine the exact P2X7R cellular expression profile.

### 3.3. P2X7R Function during Seizures and Epilepsy

As mentioned before, we now have compelling evidence suggesting that P2X7R signalling impacts both seizures and the epilepsy phenotype. Most of these data have been generated via the use of chemoconvulsants in rodent models, such as the IAKA mouse model of SE [117], the i.p. pilocarpine model [118] or the use of the non-competitive GABA_A_ receptor antagonist Pentylenetetrazol (PTZ) [119].

Several studies have demonstrated that blocking or deleting the P2X7R can have anti-convulsive effects during induced seizures [65,112,120,121]. For instance, in the IAKA mouse model of SE, seizure suppression was achieved using P2X7R KO mice, as well as through treatments with the P2X7R antagonists Brilliant Blue G (BBG) or A438079, and P2X7R-targeting antibodies delivered intra-ventricularly [65,112]. Additionally, in rats, seizures induced by intramuscular coriaria lactone were suppressed with treatments of BBG and A740003 [121]. Furthermore, in the 6 Hz test for focal seizures in mice, BBG was also found to reduce seizures [120]. Others have, however, reported either no effect [78,122] or, in contrast, a proconvulsant function when blocking P2X7Rs [116,123]. The first systematic study by Fischer et al. showed no effect of four CNS-permeable P2X7R antagonists (JNJ-47965567, AFC-5128, BBG, and tanshinone) on acute seizures using the maximal electroshock seizure threshold test (MES-T) and the subcutaneous (s.c.) PTZ seizure threshold test (PTZ-T) in mice [78]. Moreover, no effect was observed using an inbred strain of rats with genetic absence epilepsy (WAG/Rij) via the P2X7R antagonist A438079 by Dogan et al. [122]. Proconvulsive effects have been reported by Kim et al. and Rozmer et al., showing increased seizure severity during i.p. pilocarpine-induced SE in mice using P2X7R KO mice and the P2X7R antagonists OxATP, A438079, and A740003 [123], and i.p. pilocarpine in rats using the P2X7R antagonists AZ10606120 and BBG [116].

More consistent data have been observed during epilepsy with P2X7R antagonism providing generally beneficial effects. In one of the first studies, Jimenez-Pacheco et al. showed less frequent seizures in mice where epilepsy was induced via IAKA, using the P2X7R antagonist JNJ-47965567 [65]. Remarkably, seizure suppression was still evident even 7 days after drug withdrawal, suggesting some degree of disease modification. These findings were recently replicated in the same epilepsy mouse model (i.e., IAKA model) using the P2X7R antagonist JNJ-54175446 [66]. Of note, in the same study, the authors showed that treatment with the ASM levetiracetam had no effect on seizures [66]. This was similar to a previous study showing the IAKA model to be resistant to the ASM carbamazepine [124]. Meanwhile, Amhaoul et al., by using the multiple low-dose i.p. KA model in rats, showed that, while treatment with the P2X7R antagonist JNJ-47965567 did not reduce the total numbers of seizures in their model, epileptic rats experienced less severe seizures during treatment [125]. Further evidence of the antiepileptic effects of P2X7R antagonism stems from the study by Fischer et al., where treatment with the P2X7R antagonists AFC-5128- and JNJ-47965567 indicated a long-lasting significant delay in kindling development in rats using the i.p. PTZ kindling model [78]. Similarly, using the same model, Soni et al. reported reduced kindling development using the P2X7R antagonist BBG [126]. By silencing P2X7R expression via *P2rx7*-targeting siRNA, Amorim et al. showed a delayed seizure onset and seizure numbers during chronic epilepsy in the i.p. pilocarpine model in rats [127]. Therefore, while P2X7R antagonism has shown incoherent results when applied acutely, according to the model used, P2X7R antagonism during epi- lepsy seems to be generally seizure-suppressive.

In order to establish the mechanisms of how P2X7Rs contribute to seizures and epilepsy development, we recently generated mice with a KO of P2X7R in microglia or neurons using the Cre-LoxP system [62]. This showed that mice with deleted *P2rx7* in microglia displayed less severe induced seizures (IAKA and PTZ) and developed a milder form of epilepsy following IAKA-induced SE. Microglia with a *P2rx7* deletion also displayed a more anti-inflammatory phenotype when compared to mice expressing P2X7Rs in microglia post-IAKA. In contrast, mice lacking *P2rx7* in neurons showed a more severe seizure phenotype following IAKA. Of note, the overexpression of P2X7Rs in GABAergic interneurons reduced the seizure severity (IAKA and PTZ) and suppressed epileptic seizures (post-IAKA). Notably, P2X7R overexpression in GABAergic interneurons also reduced seizures in mice lacking *Scn1a*, a model of Dravet syndrome, which is a severe form of epilepsy caused by the impaired function of inhibitory interneurons [128]. While these results are in line with an increasing amount of data suggesting that the proconvulsant function of P2X7Rs is due to its function in microglia driving proinflammatory signalling, the fact that mice deficient in P2X7R in neurons develop a more severe seizure phenotype may also explain the opposing findings in different seizure models where both pro- and anti-convulsive effects of P2X7R antagonism have been reported (e.g., pilocarpine vs IAKA model) [111,116,123].

Further supporting a proconvulsant function of the P2X7R via its effects on inflammation [22] includes its upregulation in microglia following SE and during epilepsy, as observed in all models analysed. Moreover, P2X7R antagonism reduced seizure-induced inflammation including a reduction in microglia numbers and the activation status, and reduced levels of proinflammatory signalling molecules such as IL-1β [61,65,111,121,129,130,131]. Of note, IL-1β has been shown to reduce the anticonvulsant action of benzodiazepine midazolam in primary murine cortical neuron cultures [132]. Moreover, IL-1β receptor blockers such as anakinra protected the brain from SE-induced brain damage and ameliorated SE-induced epileptogenesis in mice [133,134,135]. Anakinra has also shown seizure-suppressive potential in patients with drug-refractory SE and epilepsy [75,135]. More direct evidence suggesting that P2X7R-induced inflammation contributes to its proconvulsant function stems from studies showing the absence of the antiepileptic effects of P2X7R antagonists in the presence of the anti-inflammatory drug minocycline. This was shown by Smith et al. in a model of hypoxia-induced seizures in neonatal mice [61] and by Beamer et al. in mice overexpressing the P2X7R subjected to IAKA [131]. At this point, it is noteworthy to mention that while microglia had generally been ascribed a mainly proconvulsant role, data also suggest a protective function of microglia with the depletion of microglia leading to a decreased seizure threshold [136]. Meanwhile, a study carried out by Masuch et al. showed that P2X7Rs expressed on microglia protected against N-methyl-D-aspartate (NMDA)-induced excitotoxicity in organotypic hippocampal slice cultures [137]. Whether P2X7R-based treatments, if mediated mainly via their effects on inflammation, are superior to anti-inflammatory drugs such as minocycline has not been investigated to date. Treatment with minocycline has, however, shown anticonvulsant effects in animal models and patients, and clinical trials are already underway (e.g., [138,139]). A possible advantage of drugs targeting the P2X7R, compared to other anti-inflammatory drugs, is the fact that P2X7Rs are activated mainly under pathological conditions of high eATP concentrations; therefore, P2X7R-based treatments may cause fewer adverse side effects.

With regards to the anticonvulsive effects of neuronal P2X7R on seizures, while somewhat unexpected, previous studies have shown that P2X7Rs contribute to the release and uptake of the neurotransmitters GABA and glutamate in nerve terminals of the human epileptic neocortex and epileptic rats [140,141]. P2X7Rs also reduced brain hyperexcitability via pre-synaptic inhibition in mossy fibres [142]. One could, therefore, speculate that neuronal P2X7Rs act as a defense mechanism maintaining normal brain hyperexcitability, possibly via maintaining the glutamate/GABA balance in the brain. Once, however, inflammatory pathways are activated and P2X7R signalling increases on microglia (e.g., late phases of SE, epilepsy), P2X7R activation leads to the release of proinflammatory mediators driving seizures and epilepsy development that may outweigh P2X7R-mediated anticonvulsant effects; hence, P2X7R antagonism reduces seizures/epilepsy. It is, however, important to keep in mind that functional P2X7Rs have also been found on all other major cell types within the CNS besides microglia and neurons. For example, P2X7Rs have been shown to be functional on astrocytes and oligodendrocytes [143,144], with both cell types increasingly recognised to play key roles during epilepsy development [145,146]. P2X7Rs are also expressed on endothelial cells and have been shown to contribute to BBB permeability [147]. Considering the opposing effects that microglial and neuronal P2X7Rs have on seizures, it is now crucial to further establish the exact cell type-specific function of the P2X7R during epilepsy.

### 3.4. P2X7R as an Adjunctive Treatment Target for Drug-Refractory Epilepsy

While monotherapy, referring to the use of a single medication for the treatment of a condition, is effective for most people with epilepsy, adjunctive therapy may be needed to obtain better seizure control, in particular, in patients not responding to commonly used ASMs [148]. Epilepsy is a complex disease associated with large-scale changes in gene expression, protein translation, and pathological processes, such as selective neuronal loss, gliosis, and synaptic remodelling [9,149,150,151]. As mentioned previously, targeting single genes is, therefore, unlikely to be sufficient to disrupt the development of epilepsy or have an impact on epilepsy itself and approaches that target several aspects of the underlying pathologies are most likely required. Thus, adjunctive therapies may be used to target different pathological mechanisms simultaneously (e.g., inflammation and neurotransmission) or to potentiate the effectiveness of ASMs via blocking pathological pathways interfering with their anti-seizure functions.

Providing the rationale for blocking P2X7R function on microglia as an add-on therapy for drug-refractory epilepsy, a recent study carried out by our group showed that mice overexpressing P2X7Rs in microglia exhibit a reduced responsiveness to several ASMs during IAKA-induced SE such as lorazepam, midazolam, phenytoin, and carbamazepine [131]. Notably, treatment with the anti-inflammatory drug minocycline restored normal responses to anticonvulsants in the same P2X7R-overexpressing mice, suggesting P2X7R-driven inflammation contributing to drug-refractoriness. Moreover, in line with the reduced drug responsiveness to ASMs caused by P2X7R-induced inflammation, wild-type mice subjected to IAKA exhibited decreased responsiveness to anticonvulsants when pre-treated with inflammation-inducing Lipopolysaccharides (LPS). In contrast, mice with a P2X7R deletion or those treated with the P2X7R antagonists AFC-5128 or ITH15004 and pre-treated with LPS displayed normal responses to anticonvulsants during IAKA-induced SE [131].

The first evidence demonstrating the potential of P2X7R antagonism as an adjunctive treatment for drug-refractory SE was provided in a study published in 2012 [111]. Here, mice subjected to IAKA were treated with the anticonvulsant lorazepam at a time point when responsiveness to lorazepam was already diminished (i.e., 60 min post-IAKA, which is 20 min later than the normal treatment regime in this model [152]). While neither lorazepam nor the P2X7R antagonist A438079 were able to significantly reduce seizure severity on their own, the combination of lorazepam with A438079 reduced seizure severity in all treated mice. Similarly, Fischer et al. showed in further studies that, while P2X7R antagonism had no or minimal effects in the MES-T in mice on its own, the combined use of the P2X7R antagonists JNJ-47965567 or AFC-5128 significantly improved the anticonvulsant action of carbamazepine [78]. While not tested in combination with ASMs, two additional studies investigated whether combination treatments including P2X7R-targeting drugs provide synergistic effects on seizure suppression. Here, Soni et al. reported that, by using the PTZ kindling model in rats, while the P2X7R antagonists BBG had only modest effects on the kindling score, the combined treatment of BBG with the beta-lactam antibiotic ceftriaxone, known to upregulate the expression of the Glutamate transporter 1 (GLT-1) [153], had a synergistic effect on seizure suppression, similar to the effects provided via benzodiazepine diazepam. These synergistic effects were not restricted to seizures and included other parameters including the improvement of the motor performance assessed via the rotarod test and cognitive deficits as measured via the Morris water maze [126]. In another study, Jamali-Raeufi et al., by using a rat model of intrahippocampal KA, showed a synergistic protective effect when using the P2X7R antagonist BBG and linagliptin, an inhibitor of the enzyme dipeptidyl peptidase 4 (DPP-4), previously shown to be involved during seizures [154], on hippocampal DNA fragmentation, astrogliosis as well as cognitive disturbances. However, no synergistic effects were seen on neuronal survival and seizure severity [155].

In summary, while results of treatments with P2X7R antagonists alone have been mixed, the effects of P2X7R antagonists when given in combination with ASMs seem to be more consistent, providing better seizure control.

## 4. Challenges, Considerations, and Future Perspectives

After more than a decade of extensive research on the role of the P2X7R in epilepsy, we have accumulated a large body of evidence not only regarding its involvement during seizures, but also on the therapeutic potential of targeting this receptor. The next logical steps are to identify ways of how to translate P2X7R-based treatments into clinical use. Here, the targeting of P2X7Rs in microglia as an add-on therapy for epilepsy represents a particularly promising strategy. Reducing P2X7R function in microglia, thereby dampening neuroinflammation, may in turn re-establish the anticonvulsive actions of ASMs and provide disease-modifying effects (Figure 1). Interestingly, P2X7R antagonists are already past the clinical trial stage for non-epilepsy CNS indications (e.g., depression [156]), which should help to accelerate their implementation into the clinic.

However, while we have now gathered compelling evidence of its beneficial effects in seizure suppression, including in mouse models of drug-refractory epilepsy (e.g., IAKA mouse model [124]), there are several open questions which need to be addressed:(i)Foremost, we need a clearer picture of how P2X7R down-stream signalling affects seizures. Even though the P2X7R-dependent activation of inflammation seems to be the most plausible mechanism of how P2X7Rs contribute to seizures and drug-refractoriness [62,131], the P2X7R is involved in numerous other (patho)physiological processes. For instance, the large C-terminus of P2X7Rs can interact with more than 50 cytosolic proteins involved in different physiological and pathophysiological mechanisms. Moreover, the recent discovery of opposing functions of the P2X7R during seizures with microglia P2X7Rs contributing to seizures and neuronal P2X7Rs reducing seizures [62] further demonstrates the complex P2X7R signalling cascades during epilepsy. Therefore, the identification of the cell type-specific P2X7R functions and mechanism during seizures and epilepsy will not only allow us to design the best treatment strategy, but also help to predict potential side-effects.(ii)While the data have shown increased eATP concentrations during seizures/epilepsy, we still do not know how, when, and where ATP is released during seizures, which is critical to predict P2X7R activation. Interestingly, as mentioned before, P2X7Rs themselves can contribute to ATP release. In addition, several endogenous-positive allosteric modulators such as phosphoinositide, lysophosphatidylcholine, nicotinamide, and adenine dinucleotide may sensitise the P2X7R to lower eATP concentrations [36]. While ATP availability is an important factor to take into account when considering P2X7R activation during seizures, P2X7R activity may also be regulated via the expression of different splice variants [157], which exhibit modified functions, or its subcellular redistribution (e.g., trafficking, cell surface expression changes, as reported following i.p. pilocarpine-induced SE in rats [158]).(iii)It is increasingly recognised that effective treatments require the identification of companion biomarkers. In this line, recent research has shown that P2X7R radioli-gand uptake measured via positron emission tomography (PET) imaging increased in the brain and correlated with seizure severity during IAKA-induced SE and the SE-induced underlying pathology [159,160], representing a possible readout of seizure-induced neuropathology. In addition, the P2X7R protein levels have been reported to be increased in patients with TLE and several inflammatory markers have been shown to be altered in the blood of mice post-SE in a P2X7R-dependent manner [161]. Future studies should identify biomarkers not only to predict P2X7R activation, but also to identify patients most likely benefiting from P2X7R-based treatments (e.g., P2X7R activation in the microglia).(iv)While combination treatments have been shown to be effective in SE, future research should determine their potential in reducing seizures during drug-refractory epilepsy. Future studies should also determine whether P2X7R antagonists potentiate the effects of only a certain subclass of ASMs and whether there are undesired drug interactions between P2X7R antagonists and ASMs. P2X7R function/cell-specific localisation (e.g., microglia vs. neurons) may change during disease progression, which, in turn, may impact the treatment effectiveness.(v)Inflammation is a well-accepted pathomechanism in age-associated disorders including epilepsy [162]. Notably, P2X7Rs are increasingly recognised as a treatment target for an array of age-related diseases of the CNS (e.g., neurodegenerative diseases Alzheimer’s and Parkinson’s, cancers, arthritis), possibly contributing to the chronic low-level systemic inflammation observed in these diseases [64,163]. Future studies should, therefore, be designed to establish whether chronic P2X7R activation contributes to sustained brain inflammation during ageing and whether this contributes to the observed increase in epilepsy prevalence among the aged population [164].(vi)The human *P2RX7* gene is highly polymorphic with more than 150 non-synonymous single-nucleotide polymorphisms (SNPs) reported either as loss- or gain-of-function variants (e.g., ion channel activity, pore function) and agonist-binding affinities [165], which should be taken into account when designing P2X7R-targeting drugs. Interestingly, the P2X7R rs208294 His155Tyr polymorphism has been associated with childhood febrile seizure susceptibility [166], suggesting that P2X7R polymorphisms have an impact on brain hyperexcitability. Interestingly, several P2X7R polymorphisms have been associated with common comorbidities associated with epilepsy (e.g., depression, anxiety) [25]. Future studies, should be designed to establish the impact of certain P2X7R SNPs on one’s predisposition to develop epilepsy, responsiveness to ASMs, and the presence of epilepsy-associated comorbidities.(vii)While P2X7Rs are believed to be mainly activated following tissue injury, acting as damage-sensing receptors, P2X7Rs have been ascribed numerous functions during normal physiology, e.g., the stimulation of glutamate and GABA release from astrocytes and neurons as well as the microglial release of neurotropic factors that may be inhibited by P2X7R blockade. P2X7Rs have a well-established role in the immune-inflammatory system. Would P2X7R antagonism increase the risk of infectious diseases? Importantly, P2X7Rs are expressed throughout the body (e.g., heart, gastrointestinal system [167]), thus effects of P2X7R antagonism may not be restricted to the CNS, but involve other organs. While the risk of peripheral side-effects should be minimised by using CNS-permeable P2X7R antagonists, future studies should carefully determine P2X7R safety profiles during treatments.(viii)Finally, the possible systemic side-effects caused by P2X7R antagonists and the opposing effects of the P2X7R in different cell types [62] suggest that a focal/cell type-specific delivery may provide a safer and more efficient treatment strategy. In this regard, CNS-directed gene therapy targeting the appropriate cell type(s) (e.g., overexpression of P2X7R siRNA in microglia) in target tissue(s) represents a powerful tool to achieve the long-term corrections of disorders following a single treatment [168]. Nanoparticles delivered to specific cell types or brain regions represent another potential approach. Nanostructures can be utilised as delivery agents by encapsulating drugs or attaching therapeutic drugs (e.g., siRNA) and delivering them to target tissues more precisely with a controlled release, including microglia [169,170]. Critically, epilepsy treatment offers the unique opportunity for local drug delivery directed into the seizure focus, thereby avoiding systemic drug delivery and maximising drug effects without the need for additional invasive surgery.

Nevertheless, regardless of several challenges discussed above, with its proven effects on ASMs in several pre-clinical seizure and epilepsy models and its disease-modifying potential, the P2X7R represents a promising new treatment target for drug-refractory epilepsy. Therefore, it can be hoped to see the first clinical trials using P2X7R antagonists as add-on treatments to standard ASMs in the near future.

## Figures and Tables

**Figure 1 ijms-25-06894-f001:**
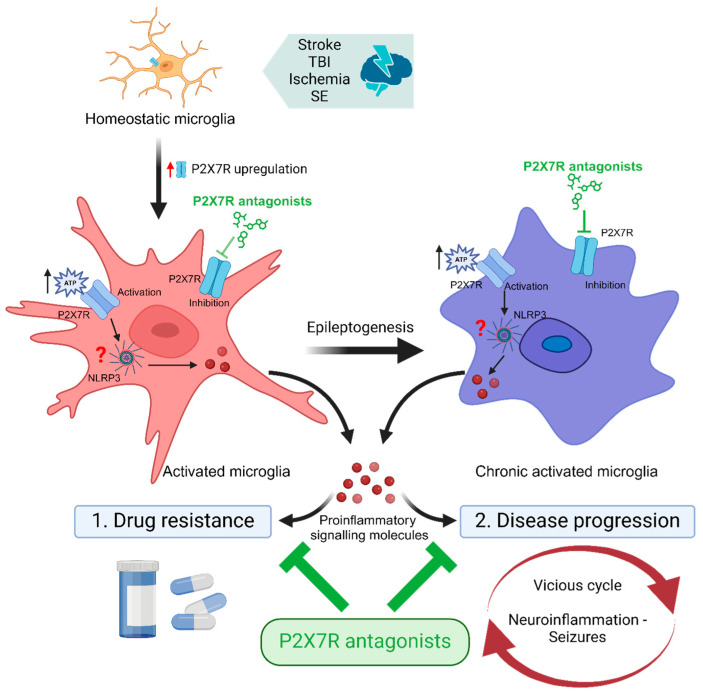
P2X7R antagonists as adjunctive treatment for epilepsy. Brain injury (e.g., stroke, traumatic brain injury [TBI], ischemia, status epilepticus [SE]) leads to the activation of microglia and an increase in microglial P2X7R expression. Upon activation via extracellularly released ATP, P2X7Rs upregulated on activated and chronic-activated microglia contribute to neuroinflammation via the promotion of the release of proinflammatory signalling molecules such as interleukin-1β. Neuroinflammation, in turn, (1) interferes with the anticonvulsive activity of anti-seizure medications (ASMs) and (2) contributes to disease progression via promoting the vicious cycle of increased neuroinflammation, hyperexcitability, and seizures. Therefore, by reducing proinflammatory signalling in the brain, add-on treatment with P2X7R antagonists may potentiate the efficacy of ASMs, reduce drug resistance, and put a break on the vicious cycle of neuroinflammation and seizures, thereby providing disease-modifying effects. Abbreviations: NLRP3, NLR family pyrin-domain-containing 3; SE, Status epilepticus; TBI, Traumatic brain injury. Created with BioRender.com.

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
