# Peer review of "The Purinergic P2X7 Receptor as a Target for Adjunctive Treatment for Drug-Refractory Epilepsy"

_ijms, 2024, doi:10.3390/ijms25136894_

Round 1

Reviewer 1 Report

Comments and Suggestions for Authors

The P2X7 receptor is a trimeric ion channel gated by extracellular ATP (eATP). The eATP-P2X7 receptor axis induces a wide variety of signaling responses in a cell type-specific manner and mediates numerous physiological functions in areas like immunology, neurology, and cardiology. Given its importance in health and disease, the P2X7 receptor represents an attractive therapeutic target for various human conditions. However, for developing targeted and efficient treatments, it seems important to get a better understanding of the cell-type differences in P2X7 expression and functions. Moreover, intracellular signaling pathways triggered by P2X7 receptors may be highly variable in different cell types and pathologies. In the context of this review, the authors focus their discussion on the function of P2X7 during epilepsy and how drugs targeting the P2X7 receptor might be used to treat unresponsiveness to current anti-seizure medications. However, the cellular localization and functions of P2X7 receptors in the nervous system remains controversial.

 Specific comments

1) I suggest to replace ATP by eATP throughout the text.

 2) Page 3, lines 139-142 “With extracellular ATP concentrations believed to be minimal during physiological conditions, P2X7R activation is suggested to occur mainly under pathological conditions ……”. This sentence seems to indicate that the stimulation of P2X7 receptors by eATP can mainly occur during pathological conditions when cell damage or inflammation results in an ATP-rich extracellular milieu. This statement needs further discussion. Indeed, various studies have shown that 1) P2X7 receptors are involved in the regulation of physiological functions, such as the development of the nervous system, neurotransmitter release, memory, and cognition to mention only these; 2) ATP can be released into the extracellular space via transporters, channels such as pannexin-1 or connexin hemichannels, the P2X7 receptor itself or through vesicular release from any cell capable of stimulated or constitutive exocytosis such as neuron, platelets or immune cells. 

3) The authors need to discuss more extensively the fact that the P2X7 receptor is considered a “silent receptor” under physiological conditions and that, therefore, undesirable side effects from P2X7R-based treatments should be mild (page 3, lines 143-145 and page 6, lines 277-280). This seems contradictory with the numerous cellular responses associated with the stimulation of P2X7 by eATP such as cellular efflux of potassium, current facilitation, macropore formation and the ectodomain shedding of many cell surface proteins by metalloproteases. 

4). The authors did not comment on the role of adenosine in epilepsy as an endogenous anticonvulsant, which can play an important role in seizure initiation, propagation and arrest. A paragraph should be added on ectonucleotidases, as they play a key role in eATP signaling. Indeed, the level of ATP in the extracellular milieu is tightly controlled by various ectonucleotidases and alkaline phosphatases expressed at the plasma membrane of numerous cells. Moreover, adenosine, generated from the hydrolysis of eATP, is a potent regulator of inflammation.

Author Response

We would like to thank the reviewer for the constructive suggestions to our manuscript. Please see below a point-by-point discussion of the concerns raised.

1) I suggest to replace ATP by eATP throughout the text.

This has been amended according to the reviewer’s suggestion.

2) Page 3, lines 139-142 “With extracellular ATP concentrations believed to be minimal during physiological conditions, P2X7R activation is suggested to occur mainly under pathological conditions ……”. This sentence seems to indicate that the stimulation of P2X7 receptors by eATP can mainly occur during pathological conditions when cell damage or inflammation results in an ATP-rich extracellular milieu. This statement needs further discussion. Indeed, various studies have shown that 1) P2X7 receptors are involved in the regulation of physiological functions, such as the development of the nervous system, neurotransmitter release, memory, and cognition to mention only these; 2) ATP can be released into the extracellular space via transporters, channels such as pannexin-1 or connexin hemichannels, the P2X7 receptor itself or through vesicular release from any cell capable of stimulated or constitutive exocytosis such as neuron, platelets or immune cells. 

According to the reviewer’s suggestions, we have included a paragraph on the role of the P2X7R during physiological conditions. We have also included a short paragraph on what is known regarding ATP release during seizures and epilepsy.

Line 129: “Consequently, it was proposed that ATP may act as a danger signal in the brain and P2X7Rs function as danger-sensors contributing to the progression of brain diseases [54]. On the other hand, regarding its wide expression in various mammalian cells, it can be suggested that this receptor may have a role in basal cell signalling in physiological functions and homeostasis [36]. Therefore, it is important to note that P2X7Rs, beside its tissue damage-activated actions, also carry out multiple functions during physiological conditions, which should be considered when evaluating safety profiles of P2X7R-based therapies. This includes its effects on CNS development, synaptic plasticity and memory, neurogenesis, neurotransmitter release, and altered immune responses in P2X7R knock-out (KO) mice [55-59]. Moreover, further suggesting a physiological role of P2X7Rs in the CNS, we have shown altered mRNA and microRNA expression profiles during physiological conditions in hippocampal tissue from P2X7R KO mice [60-62]. It is, however, noteworthy to mention that P2X7R KO mice are viable and show protection against most noxious insults [63, 64]. Importantly, as discussed within the following sections, P2X7R-based treatments may provide some degree of disease-modification [65, 66], reducing time of treatment and, thereby, the potential negative impact on normal brain function(s).

Line 186: 3.1 “3.1 ATP release during seizures and epilepsy

eATP is one of the primary damage-associated molecular pattern (DAMP) to be released at sites of tissue injury serving as a physiological protective mechanism [54, 85]. eATP concentrations are tightly regulated via different ATP release mechanisms and eATP-degrading ectoenzymes [86, 87]. In the CNS, eATP has been described as a cotransmitter with glutamate [88, 89] and GABA [90, 91]. ATP has also been shown to be released from non-neuronal cells such as astrocytes [92]. ATP release mechanisms include exocytotic and non-exocytotic mechanisms involving the Cl--dependent vesicular nucleotide transporter (VNUT) [93], voltage-dependent anion channels [94], ATP-binding cassette transporters [95], the purinergic P2X7R and hemichannels such as connexins and pannexins [96-98]. Due to its huge concentration gradient, ATP has also been shown to be released passively through damaged cell membranes [54]. Once released, eATP is rapidly broken down via ectonucleotidases into different breakdown products (e.g., ADP, adenosine), signalling molecules in their own right [31]. Demonstrating altered concentrations during seizures, eATP concentrations have been found increased in the brain of a seizure-prone strain of mice [inbred DBA/2 (D2)] [99] and in slices of rat hippocampi stimulated via depolarising high K+ concentrations [100]. Using a rat model where SE was induced via an intraperitoneal injection of pilocarpine, Dona et al., reported increased extracellular levels of ADP, adenosine monophosphate (AMP), and adenosine. Purine levels, including eATP, were also found to be increased in the brain of mice following an epileptic seizure [51]. Regarding the ATP release mechanisms during seizures, most research has focused on the pannexin-1 hemichannel. Increases in eATP have been shown to be mediated via pannexin-1 in rat hippocampal slices [101]. More recent studies have shown that blocking Pannexin-1 reduced eATP in resected tissue from patients with epilepsy. Interestingly, pharmacological inhibition of Pannexin-1 also blocked ictal discharges in human cortical brain tissue slices and mice KO for pannexin-1 had a milder KA-induced SE phenotype [102], suggesting ATP release mechanisms as potential target for seizure control. As previously mentioned, it is, however, important to keep in mind that, once released, eATP is rap-idly broken down via the action of several ectonucleotidases into different breakdown products including adenosine [87]. Thus, while activating ATP-sensitive P2X receptors, eATP also contributes to the extracellular adenosine pool, impacting thereby further on the seizure threshold, which will be briefly discussed in the next paragraph [103].”

3) The authors need to discuss more extensively the fact that the P2X7 receptor is considered a “silent receptor” under physiological conditions and that, therefore, undesirable side effects from P2X7R-based treatments should be mild (page 3, lines 143-145 and page 6, lines 277-280). This seems contradictory with the numerous cellular responses associated with the stimulation of P2X7 by eATP such as cellular efflux of potassium, current facilitation, macropore formation and the ectodomain shedding of many cell surface proteins by metalloproteases. 

This has been included as suggested by the reviewer:

See above and Line 360: A possible advantage of drugs targeting the P2X7R, compared to other anti-inflammatory drugs, is the fact that P2X7Rs are activated mainly under pathological conditions of high eATP concentrations, therefore, P2X7R-based treatments may cause fewer adverse side effects.”

Please also see:

Line 521: (vii) While P2X7Rs are believed to be mainly activated following tissue injury acting as damage-sensing receptor, P2X7Rs have been ascribed numerous functions during normal physiology, e.g. stimulation of glutamate and GABA release from astrocytes as well as microglial release of neurotropic factors that may be inhibited by P2X7R blockade. P2X7Rs have a well-established role in the immune-inflammatory system. Would P2X7R antagonism increase the risk of infectious diseases? Importantly, P2X7Rs are expressed throughout the body (e.g. heart, gastrointestinal system [167]), thus effects of P2X7R antagonism may not be restricted to the CNS but involve other organs. While the risk of peripheral side effects should be minimized by using CNS-permeable P2X7R antagonists, future studies should carefully determine P2X7R safety profiles during treatments.”

4) The authors did not comment on the role of adenosine in epilepsy as an endogenous anticonvulsant, which can play an important role in seizure initiation, propagation and arrest. A paragraph should be added on ectonucleotidases, as they play a key role in eATP signaling. Indeed, the level of ATP in the extracellular milieu is tightly controlled by various ectonucleotidases and alkaline phosphatases expressed at the plasma membrane of numerous cells. Moreover, adenosine, generated from the hydrolysis of eATP, is a potent regulator of inflammation.

According to the reviewer’s suggestion, we have included a short paragraph on the role of adenosine.

Line 219: “Adenosine is formed by dephosphorylation of eATP from neurons and astrocytes via the membrane-bound ecto 5ʹ-nucleotidases CD39 and CD73, and transported through the lipid bilayer via nucleoside transporters in the extracellular space [86] where it is largely under control of the enzyme adenosine kinase (ADK). Adenosine is now a well characterized anti-inflammatory endogenous anticonvulsant of the brain and mediator of seizure arrest [104-107]. A strong increase of extracellular adenosine during seizure activity in the hippocampi of patients was documented and considered to be sufficient to terminate ongoing seizure activity. On the other hand, dysregulation of adenosine signalling (receptors, transporters, upregulation of ADK as well as DNA hypermethylation) is involved in the development of epilepsy [103]. Several studies have also reported changes in the density or affinity of the four adenosine receptors in epileptic models of rodents as well as in human epileptic tissues, but this needs further clarification. In animal models, stimulation of the A1 (and A2A) receptor with brain permeable agonists effectively contributes to seizure suppression. Of note, a systemic administration of adenosine itself is not possible due to strong cardiovascular side effects [108]. In conclusion, treatments that facilitate the adenosine signalling system (adenosine augmentation therapies) emerges as a rational therapeutic target with the potential to suppress seizures, but also to prevent epileptogenesis [8]. Of note, recently, intracerebral cell therapy approaches have been developed for long-term therapeutic adenosine delivery at sites of injury or pathology [8, 103].

Reviewer 2 Report

Comments and Suggestions for Authors

This manuscript provides details about the purinergic P2X7 with potential as adjunctive treatment target for drug-refractory epilepsy, which can be a good summary into the clinical value of P2X7 in healthcare. Overall, this manuscript was written well. However, the manuscript can be improved further. Bellows are my comments. 

1. Abstract can be more concise to only provide the highlights of this manuscript. Current abstract is a little tedious. For example, lines 14-19, it can be rephrased to make it more summarized. Lines 20-32, many sentences can also be further trimmed.

2. For introduction part, lines 36-63 mainly introduced what is epilepsy.  I think too many contents are included here because the part of knowledge is well known, which does not help make this manuscript unique.  Lines 82-96 talk about the drug refractoriness. But I think lines 88-96 can be revised to be more concise.

3. Line 104 indicates that the focus of this review is discussing “how drugs targeting the P2X7R may be used in a clinically-relevant setting “, but line 117 indicates that this review “will focus on the purinergic P2X7R and its role during epilepsy”. I think these two themes are different because the former highlights the clinical insights while the latter mainly focuses on basic physiological functions. Can authors revise to make the focus of this review more specific? 

4. Lines 121-144 have different font style from other paragraphs. Additionally, authors use such a long paragraph to talk about what is P2X7R, and I do not see the necessity. Short and concise introduction is sufficient from my perspectives. 

5. Lines 165-170 also mentioned other neurodegenerative diseases, I think these introductions can be deleted because this review focused on epilepsy.

6. Lines 213-250 cited much research about effects of P2X7R on epilepsy. After reading this paragraph, an obvious feeling is that authors just are describing the published results without viewpoints of authors. A good review is not to just describe the published results. 

Author Response

We would like to thank the reviewer very much for the suggestions. Please find below a point-by-point response to the concerns raised.

  1. Abstract can be more concise to only provide the highlights of this manuscript. Current abstract is a little tedious. For example, lines 14-19, it can be rephrased to make it more summarized. Lines 20-32, many sentences can also be further trimmed.

The abstract has been shortened according to the reviewer’s recommendations.

  1. For introduction part, lines 36-63 mainly introduced what is epilepsy.  I think too many contents are included here because the part of knowledge is well known, which does not help make this manuscript unique.  Lines 82-96 talk about the drug refractoriness. But I think lines 88-96 can be revised to be more concise.

The mentioned paragraphs have been shortened accordingly.

  1. Line 104 indicates that the focus of this review is discussing “how drugs targeting the P2X7R may be used in a clinically-relevant setting “, but line 117 indicates that this review “will focus on the purinergic P2X7R and its role during epilepsy”. I think these two themes are different because the former highlights the clinical insights while the latter mainly focuses on basic physiological functions. Can authors revise to make the focus of this review more specific? 

Thank you for pointing this out. In our revised manuscript we have removed the sentence “will focus on the purinergic P2X7R and its role during epilepsy”.

  1. Lines 121-144 have different font style from other paragraphs. Additionally, authors use such a long paragraph to talk about what is P2X7R, and I do not see the necessity. Short and concise introduction is sufficient from my perspectives. 

Thank you for spotting this - styles have been adjusted accordingly.

Regarding the description of the P2X7R. We would like to keep the detailed description of the P2X7R as this review is not only targeted to researchers in the field of purinergic signalling but also to researchers in the field of epilepsy and other brain diseases. This would also be in line with reviewer 1 and 2 who requested a more detailed description on the P2X7R.

  1. Lines 165-170 also mentioned other neurodegenerative diseases, I think these introductions can be deleted because this review focused on epilepsy.

While we would like to keep the reference to other brain diseases, we agree with the reviewer and have shortened this paragraph accordingly.

Line 170: As a consequence, the P2X7R has been linked to a plethora of brain diseases ranging from acute conditions such as a stroke to chronic conditions including neurodegenerative diseases, psychiatric conditions [63, 64, 84], and, as discussed within this review, epilepsy [24, 25].”

  1. Lines 213-250 cited much research about effects of P2X7R on epilepsy. After reading this paragraph, an obvious feeling is that authors just are describing the published results without viewpoints of authors. A good review is not to just describe the published results. 

According to the reviewer’s suggestions, we have included several new paragraphs in our revised manuscript where we speculate on the possible implications of reported findings:

Line 365: “With regards to the anticonvulsive effects of neuronal P2X7R on seizures, while somewhat unexpected, previous studies have shown that P2X7Rs contribute to the release and uptake of the neurotransmitters GABA and glutamate in nerve terminals of the human epileptic neocortex and epileptic rats [140, 141]. P2X7Rs also reduced brain hyperexcitability via pre-synaptic inhibition in mossy fibers [142]. One could, therefore, speculate that neuronal P2X7Rs act as a defense mechanism maintaining normal brain hyperexcitability, possibly via maintaining the glutamate/GABA balance in the brain. Once, however, inflammatory pathways are activated and P2X7R signalling increases on microglia (e.g., late phases of SE, epilepsy), P2X7R activation leads to the release of pro-inflammatory mediators driving seizures and epilepsy development that may outweigh P2X7R-mediated anticonvulsant effects, hence P2X7R antagonism reduces seizures/epilepsy. It is, however, important to keep in mind that functional P2X7Rs have also been found on all other major cell types within the CNS besides microglia and neurons. For example, P2X7Rs have been shown to be functional on astrocytes and oligodendrocytes [143, 144], with both cell types increasingly recognized to play key roles during epilepsy development [145, 146]. P2X7Rs are also expressed on endothelial cells and have been shown to contribute to BBB permeability [147]. Considering the opposing effects that microglial and neuronal P2X7Rs have on seizures, it is now crucial to further establish the exact cell type-specific function of the P2X7R during epilepsy.”

Line 468: “(i) Foremost, we need a clearer picture of how P2X7R down-stream signalling affects seizures. Even though P2X7R-dependent activation of inflammation seems to be the most plausible mechanism of how P2X7Rs contribute to seizures and drug-refractoriness [62, 131], the P2X7R is involved in numerous other (patho)physiological processes. For instance, the large C-terminus of P2X7Rs can interact with more than 50 cytosolic proteins involved in different physiological and pathophysiological mechanisms. Moreover, the recent discovery of opposing functions of the P2X7R during seizures with microglia P2X7Rs contributing to seizures and neuronal P2X7Rs reducing seizures [62], further demonstrates the complex P2X7R signal-ling cascades during epilepsy. Therefore, the identification of the cell type-specific P2X7R functions and mechanism during seizures and epilepsy will not only allow us to design the best treatment strategy, but also help to predict potential side effects.

(ii) While data has shown increased eATP concentrations during seizures/epilepsy, we still do not know how, when and where ATP is released during seizures, critical to predict P2X7R activation. Interestingly, as mentioned before, P2X7Rs itself can contribute to ATP release. In addition, several endogenous positive allosteric modulators such as phosphoinositide, lysophosphatidylcholine, nicotinamide adenine dinucleotide may sensitize the P2X7R to lower eATP concentrations [36]. While ATP availability is an important factor to take into account when considering P2X7R activation during seizures, P2X7R activity may also be regulated via the expression of different splice variants [157], which exhibit modified functions, or its subcellular redistribution (e.g., trafficking, cell surface expression changes as reported post-pilocarpine induced SE in rats [158]).

(iii) It is increasingly recognized that effective treatments require the identification of companion biomarkers. In this line, recent research has shown P2X7R radioligand uptake measured via positron emission tomography (PET) imaging increased in the brain and correlated with seizure severity during IAKA-induced SE and SE-induced underlying pathology [159, 160], representing a possible readout of seizure-induced neuro-pathology. In addition, P2X7R protein levels have been reported to be increased in patients with TLE and several inflammatory markers have been shown to be altered in the blood of mice post-SE in a P2X7R-dependent manner [161]. Future studies should identify biomarkers not only to predict P2X7R activation but also to identify patients most likely benefiting from P2X7R-based treatments (e.g., P2X7R activation in microglia).

(iv) While combination treatments have been shown to be effective in SE, future research should determine its potential in reducing seizures during drug-refractory epilepsy. Future studies should also determine whether P2X7R antagonists potentiate the effects of only a certain subclass of ASMs and whether there are undesired drug inter-actions between P2X7R antagonists and ASMs. P2X7R function/cell specific localization (e.g., microglia vs neurons) may changes during disease progression, which, in turn, may impact on treatment effectiveness.

(v) Inflammation is a well-recognized pathomechanism in age-associated disorders including epilepsy [162]. Notably, P2X7Rs are increasingly recognized as treatment target for an array of age-related diseases of the CNS (e.g., neurodegenerative diseases Alzheimer’s and Parkinson’s, cancers, arthritis) possibly contributing to the chronic low-level systemic inflammation observed in these diseases [64, 163]. Future studies should, therefore, be designed to establish whether a chronic P2X7R activation contributes to a sustained brain inflammation during ageing and whether this contributes to the observed increase in epilepsy prevalence among the aged population [164].

(vi) The human P2RX7 gene is highly polymorphic with more than 150 non-synonymous single nucleotide polymorphisms (SNPs) reported either as loss- or gain-of-function variants (e.g., ion channel activity, pore function) and agonist binding affinities [165], which should be taken into account when designing P2X7R-targeting drugs. Interestingly, the P2X7R rs208294 His155Tyr polymorphism has been associate with childhood febrile seizure susceptibility [166], suggesting P2X7R polymorphisms impacting on brain hyperexcitability. Interestingly, several of P2X7R polymorphisms have been associated with common comorbidities associated with epilepsy (e.g., depression, anxiety) [25]. Future studies, should be designed to establish the impact of certain P2X7R SNPs on the predisposition to develop epilepsy, responsiveness to ASMs and presence of epilepsy-associated comorbidities.

(vii) While P2X7Rs are believed to be mainly activated following tissue injury acting as damage-sensing receptor, P2X7Rs have been ascribed numerous functions during normal physiology, e.g. stimulation of glutamate and GABA release from astrocytes as well as microglial release of neurotropic factors that may be inhibited by P2X7R blockade. P2X7Rs have a well-established role in the immune-inflammatory system. Would P2X7R antagonism increase the risk of infectious diseases? Importantly, P2X7Rs are expressed throughout the body (e.g. heart, gastrointestinal system [167]), thus effects of P2X7R antagonism may not be restricted to the CNS but involve other organs. While the risk of peripheral side effects should be minimized by using CNS-permeable P2X7R antagonists, future studies should carefully determine P2X7R safety profiles during treatments.

(viii) Finally, the possible systemic side-effects caused by P2X7R antagonists and the opposing effects of the P2X7R in different cell types [62] suggests that a focal/cell-type specific delivery may provide a safer and more efficient treatment strategy. In this regard, CNS-directed gene therapy targeting the appropriate cell type(s) (e.g., microglia overexpressing P2X7R siRNA) in target tissue(s) represent a powerful tool to achieve long-term corrections of disorders following a single treatment [168]. Nanoparticles delivered to specific cell types or brain regions represent another potential approach. Nanostructures can be utilized as delivery agents by encapsulating drugs or attaching therapeutic drugs (e.g. siRNA) and deliver them to target tissues more precisely with a controlled release, including microglia [169, 170]. Critically, epilepsy treatment offers the unique opportunity for local drug delivery directed into the seizure locus, thereby avoiding systemic drug delivery and maximizing drug effects without the need for additional invasive surgery.” 

Reviewer 3 Report

Comments and Suggestions for Authors

This review on the potential relevance of P2X7R in the context of epilepsy has the main merit to call the reader attention to the link between P2X7R and the efficiency of ASM, apart from reviewing, once more, the numerous phenomenological studies (without underlying mechanistic insights) reporting effects (or lack of thereof) of P2X7R antagonists. The following points should be considered:

1-The whole discussion of the role(s) of P2X7R in the context of epilepsy (section 3.1) cannot be dissociated from the generation of extracellular ATP, which is not even mentioned.

2- The concept that ATTP is a danger signal in the brain (Rodrigues et al., 2015, Front Neurosci 9:148) is missing and seems critically required.

3-What switches on P2X7R during epilepsy? i) upregulation of P2X7R that have increased activity? ii) increased ATP release reaching levels sufficient to activate P2X7R? iii) combined up-regulation of P2X7R and increased ATP release? iv) subcellular redistribution of P2X7R (trafficking, plasma membrane re-localization)? Some discussion seems required.

4-The use of an adequate scientific terminology is required. 'Expression' has a defined biological meaning, designating the process of gene readout and formation of mRNA. Then, the mRNA is translated into proteins, which have densities, levels or localizations, but are certainly NOT expressed. Thus, 'P2X7R expression (l.190) CANNOT be assessed using antibodies (l.191). This erroneous use of ‘expression’ needs to be corrected throughout the whole manuscript.

There is a recurrent willingness to consider the translation of P2X7R antagonists into clinics, even without knowing where they are and what they in the epileptic brain. Such a translational effort would require discussing some question, such as:

5-What is known about the cellular localization of P2X7R in the human brain and epilepsy-associated alterations?

6-P2X7R has numerous polymorphisms. Please discuss the relation of investigating if P2X7R polymorphisms are related to: i) the probability to suffer from epilepsy; ii) the evolution of epilepsy; iii) the sensitivity to ASM.

7-P2X7R are of notorious relevance to control the immune-inflammatory system. The potential side effects resulting from the eventual clinical use of P2X7R antagonists that are related to the risks of infections should be discussed.

8-Although epilepsy affects individuals of all ages, there is a surge of epilepsy in the elderly. This justifies requesting to discuss the alteration of brain P2X7R upon aging.

Author Response

We would like to thank the reviewer for the constructive feedback, thereby helping to improve our manuscript. Please find below a detailed point-by-point discussion of the concerns and suggestions raised.

1.The whole discussion of the role(s) of P2X7R in the context of epilepsy (section 3.1) cannot be dissociated from the generation of extracellular ATP, which is not even mentioned.

According to the reviewer’s suggestions we have included a short summary of what is know regarding ATP release during seizures and epilepsy.

Line 186: 3.1 “3.1 ATP release during seizures and epilepsy

eATP is one of the primary damage-associated molecular pattern (DAMP) to be released at sites of tissue injury serving as a physiological protective mechanism [54, 85]. eATP concentrations are tightly regulated via different ATP release mechanisms and eATP-degrading ectoenzymes [86, 87]. In the CNS, eATP has been described as a cotransmitter with glutamate [88, 89] and GABA [90, 91]. ATP has also been shown to be released from non-neuronal cells such as astrocytes [92]. ATP release mechanisms include exocytotic and non-exocytotic mechanisms involving the Cl--dependent vesicular nucleotide transporter (VNUT) [93], voltage-dependent anion channels [94], ATP-binding cassette transporters [95], the purinergic P2X7R and hemichannels such as connexins and pannexins [96-98]. Due to its huge concentration gradient, ATP has also been shown to be released passively through damaged cell membranes [54]. Once released, eATP is rapidly broken down via ectonucleotidases into different breakdown products (e.g., ADP, adenosine), signalling molecules in their own right [31]. Demonstrating altered concentrations during seizures, eATP concentrations have been found increased in the brain of a seizure-prone strain of mice [inbred DBA/2 (D2)] [99] and in slices of rat hippocampi stimulated via depolarising high K+ concentrations [100]. Using a rat model where SE was induced via an intraperitoneal injection of pilocarpine, Dona et al., reported increased extracellular levels of ADP, adenosine monophosphate (AMP), and adenosine. Purine levels, including eATP, were also found to be increased in the brain of mice following an epileptic seizure [51]. Regarding the ATP release mechanisms during seizures, most research has focused on the pannexin-1 hemichannel. Increases in eATP have been shown to be mediated via pannexin-1 in rat hippocampal slices [101]. More recent studies have shown that blocking Pannexin-1 reduced eATP in resected tissue from patients with epilepsy. Interestingly, pharmacological inhibition of Pannexin-1 also blocked ictal discharges in human cortical brain tissue slices and mice KO for pannexin-1 had a milder KA-induced SE phenotype [102], suggesting ATP release mechanisms as potential target for seizure control. As previously mentioned, it is, however, important to keep in mind that, once released, eATP is rap-idly broken down via the action of several ectonucleotidases into different breakdown products including adenosine [87]. Thus, while activating ATP-sensitive P2X receptors, eATP also contributes to the extracellular adenosine pool, impacting thereby further on the seizure threshold, which will be briefly discussed in the next paragraph [103].”

  1. The concept that ATTP is a danger signal in the brain (Rodrigues et al., 2015, Front Neurosci 9:148) is missing and seems critically required.

This has been included accordingly.

Line 129: “Consequently, it was proposed that ATP may act as a danger signal in the brain and P2X7Rs function as danger-sensors contributing to the progression of brain diseases [54].”

  1. What switches on P2X7R during epilepsy? i) upregulation of P2X7R that have increased activity? ii) increased ATP release reaching levels sufficient to activate P2X7R? iii) combined up-regulation of P2X7R and increased ATP release? iv) subcellular redistribution of P2X7R (trafficking, plasma membrane re-localization)? Some discussion seems required.

We agree with the reviewer that these are important questions to be answered before P2X7R-based treatments can enter the clinic and has been added as suggested.

Line 468: 4. Challenges, Considerations and Future Perspectives, (i) Foremost, we need a clearer picture of how P2X7R down-stream signalling affects seizures. Even though P2X7R-dependent activation of inflammation seems to be the most plausible mechanism of how P2X7Rs contribute to seizures and drug-refractoriness [62, 131], the P2X7R is involved in numerous other (patho)physiological processes. For instance, the large C-terminus of P2X7Rs can interact with more than 50 cytosolic proteins involved in different physiological and pathophysiological mechanisms. Moreover, the recent discovery of opposing functions of the P2X7R during seizures with microglia P2X7Rs contributing to seizures and neuronal P2X7Rs reducing seizures [62], further demonstrates the complex P2X7R signalling cascades during epilepsy. Therefore, the identification of the cell type-specific P2X7R functions and mechanism during seizures and epilepsy will not only allow us to design the best treatment strategy, but also help to predict potential side effects.

(ii) While data has shown increased eATP concentrations during seizures/epilepsy, we still do not know how, when and where ATP is released during seizures, critical to predict P2X7R activation. Interestingly, as mentioned before, P2X7Rs itself can contribute to ATP release. In addition, several endogenous positive allosteric modulators such as phosphoinositide, lysophosphatidylcholine, nicotinamide adenine dinucleotide may sensitize the P2X7R to lower eATP concentrations [36]. While ATP availability is an important factor to take into account when considering P2X7R activation during seizures, P2X7R activity may also be regulated via the expression of different splice variants [157], which exhibit modified functions, or its subcellular redistribution (e.g., trafficking, cell surface expression changes as reported post-pilocarpine induced SE in rats [158]).”

  1. The use of an adequate scientific terminology is required. 'Expression' has a defined biological meaning, designating the process of gene readout and formation of mRNA. Then, the mRNA is translated into proteins, which have densities, levels or localizations, but are certainly NOT expressed. Thus, 'P2X7R expression (l.190) CANNOT be assessed using antibodies (l.191). This erroneous use of ‘expression’ needs to be corrected throughout the whole manuscript.

Thank you for pointing this out. This has been corrected throughout our revised manuscript according to the reviewer’s suggestion.

  1. There is a recurrent willingness to consider the translation of P2X7R antagonists into clinics, even without knowing where they are and what they in the epileptic brain. Such a translational effort would require discussing some question, such as:

5.1 What is known about the cellular localization of P2X7R in the human brain and epilepsy-associated alterations?

We strongly agree with the reviewer. The design of effective treatments requires a detailed knowledge of where, in what cell types and under what conditions target genes are expressed. Accordingly, we have extended the paragraph on the cell type specific localization of P2X7Rs during epilepsy including our recently published paper on the cell type specific role of P2X7Rs during seizures.

Line 259: “Using path clamp recordings, increased P2X7R-depencent currents have also been detected at neural progenitor cells (NPCs) in the subgranular zone of the dentate gyrus in hippocampal brain slices [116]. Finally, in a recent study using the Cre-LoxP system, Alves et al. showed altered responses to chemoconvulsants in mice KO for the P2X7R in microglia (P2rx7:Cx3cr1-Cre) and neurons (P2rx7:Thy-1-Cre), suggesting P2X7Rs being functional in both cell types. In the same study, the authors further showed increased P2X7R-mediated currents in GABAergic interneurons via patch clamp in brain slices from mice subjected to IAKA and elevated P2RX7 mRNA expression in excitatory and inhibitory neurons in the hippocampus of patients with TLE [62].”

5.2 P2X7R has numerous polymorphisms. Please discuss the relation of investigating if P2X7R polymorphisms are related to: i) the probability to suffer from epilepsy; ii) the evolution of epilepsy; iii) the sensitivity to ASM.

This has been included as suggested by the reviewer:

Line 515: “(vi) The human P2RX7 gene is highly polymorphic with more than 150 non-synonymous single nucleotide polymorphisms (SNPs) reported either as loss- or gain-of-function variants (e.g., ion channel activity, pore function) and agonist binding affinities [165], which should be taken into account when designing P2X7R-targeting drugs. Interestingly, the P2X7R rs208294 His155Tyr polymorphism has been associate with childhood febrile seizure susceptibility [166], suggesting P2X7R polymorphisms impacting on brain hyperexcitability. Interestingly, several of P2X7R polymorphisms have been associated with common comorbidities associated with epilepsy (e.g., depression, anxiety) [25]. Future studies, should be designed to establish the impact of certain P2X7R SNPs on the predisposition to develop epilepsy, responsiveness to ASMs and presence of epilepsy-associated comorbidities.”

5.3 P2X7R are of notorious relevance to control the immune-inflammatory system. The potential side effects resulting from the eventual clinical use of P2X7R antagonists that are related to the risks of infections should be discussed.

This has been included as suggested:

Line 526: (vii) While P2X7Rs are believed to be mainly activated following tissue injury acting as damage-sensing receptor, P2X7Rs have been ascribed numerous functions during normal physiology, e.g. stimulation of glutamate and GABA release from astrocytes as well as microglial release of neurotropic factors that may be inhibited by P2X7R blockade. P2X7Rs have a well-established role in the immune-inflammatory system. Would P2X7R antagonism increase the risk of infectious diseases? Importantly, P2X7Rs are expressed throughout the body (e.g. heart, gastrointestinal system [167]), thus effects of P2X7R antagonism may not be restricted to the CNS but involve other organs. While the risk of peripheral side effects should be minimized by using CNS-permeable P2X7R antagonists, future studies should carefully determine P2X7R safety profiles during treatments.”

5.4 Although epilepsy affects individuals of all ages, there is a surge of epilepsy in the elderly. This justifies requesting to discuss the alteration of brain P2X7R upon aging.

This has been included according to the reviewer’s suggestion.

Line 507: “(v) Inflammation is a well-recognized pathomechanism in age-associated disorders including epilepsy [162]. Notably, P2X7Rs are increasingly recognized as treatment target for an array of age-related diseases of the CNS (e.g., neurodegenerative diseases Alzheimer’s and Parkinson’s, cancers, arthritis) possibly contributing to the chronic low-level systemic inflammation observed in these diseases [64, 163]. Future studies should, therefore, be designed to establish whether a chronic P2X7R activation contributes to a sustained brain inflammation during ageing and whether this contributes to the observed increase in epilepsy prevalence among the aged population [164].”

Round 2

Reviewer 1 Report

Comments and Suggestions for Authors

All comments have been addressed

Reviewer 2 Report

Comments and Suggestions for Authors

This manuscript has been improved, and I suggest it can be accept for publication.